# Intranasal Immunization with Liposome-Displayed Receptor-Binding Domain Induces Mucosal Immunity and Protection against SARS-CoV-2

**DOI:** 10.3390/pathogens11091035

**Published:** 2022-09-12

**Authors:** Wei-Chiao Huang, Kevin Chiem, Luis Martinez-Sobrido, Jonathan F. Lovell

**Affiliations:** 1Department of Biomedical Engineering, University at Buffalo, Buffalo, NY 14260, USA; 2POP Biotechnologies, Buffalo, NY 14128, USA; 3Texas Biomedical Research Institute, San Antonio, TX 78227, USA

**Keywords:** vaccines, liposomes, SARS-CoV-2, intranasal, RBD

## Abstract

The global pandemic, caused by severe acute respiratory syndrome coronavirus 2 (SARS-CoV-2), has led to efforts in developing effective vaccine approaches. Currently, approved coronavirus disease 2019 (COVID-19) vaccines are administered through an intramuscular (I.M.) route. Here, we show that the SARS-CoV-2 spike (S) glycoprotein receptor-binding domain (RBD), when displayed on immunogenic liposomes, can be intranasally (I.N.) administered, resulting in the production of antigen-specific IgA and antigen-specific cellular responses in the lungs. Following I.N. immunization, antigen-presenting cells of the lungs took up liposomes displaying the RBD. K18 human ACE2-transgenic mice that were immunized I.M or I.N with sub-microgram doses of RBD liposomes and that were then challenged with SARS-CoV-2 had a reduced viral load in the early course of infection, with I.M. achieving complete viral clearance. Nevertheless, both vaccine administration routes led to full protection against lethal viral infection, demonstrating the potential for the further exploration and optimization of I.N immunization with liposome-displayed antigen vaccines.

## 1. Introduction

Severe acute respiratory syndrome coronavirus 2 (SARS-CoV-2) has sparked an unprecedented global health crisis [1]. The asymptomatic spread of the disease further complicates public health countermeasures [2]. SARS-CoV-2 vaccines have been a major mitigation strategy [3], and include mRNA-lipid nanoparticles [4,5,6,7], adenoviral vectors [8,9,10] that encode the spike (S) viral glycoprotein, inactivated virus [11,12,13], and subunit protein vaccines [14,15,16].

To date, approved vaccines for coronavirus disease 2019 (COVID-19) and most vaccines in development are administered through the intramuscular (I.M.) route, which produces systemic immune responses including serum IgG, and it remains a matter of controversy whether this produces meaningful mucosal immune responses [17]. SARS-CoV-2 mRNA-based vaccines, administered systemically, induce only weak mucosal immune responses [18]. Generally, prophylactic I.M. immunization produces antibody responses dominated by sera IgG, and lacks mucosal immune responses localized to the site of virus infection [19,20]. The mucosal immune response generates neutralizing antibodies against SARS-CoV-2 in the upper respiratory tract, and the absence of a mucosal immune response increases the risk of SARS-CoV-2 transmission from person to person through vaccinated people who can still be infected by the virus [21]. Intranasal immunization (I.N.) could provide protection against infection through the upper respiratory tract with a robust local IgA response, which might lead to better protection against infection.

Several studies have shown that I.N. administration can generate protective immunity in vivo [22,23]. I.N. administration using a spike (S) protein-based chimpanzee adenovirus-vectored vaccine (ChAd-SARS-CoV-2-S) induced antibody responses in serum and anti-spike IgG and IgA, secreting long-lived plasma cells in bone marrow [24], and is in a phase I clinical study (NCT04751682). [25] Hamsters vaccinated with a replication-defective, shelf-stable adenoviral type 5 (Ad5) vector vaccine candidate expressing the S protein through oral or I.N. administration showed a decreased viral load in the nose and lungs compared to non-vaccinated hamsters post-challenge with SARS-CoV-2 [26]. The same platform combining an S protein and nucleocapsid (N) protein elicited specific IgA responses in a phase 1 clinical trial (NCT04564702) [27].

The formulation or engineering of antigens into particles has attracted attention, as this can frequently induce stronger immune responses [28]. Mechanisms for this phenomenon include enhancing delivery to antigen-presenting cells (APC) and increasing antigen-specific B cell activation [29,30]. Traditional methods for antigen conjugation usually involve engineering substantial portions of the protein, or the use of bioconjugate chemistry. Our alternative approach makes use of liposomes containing cobalt porphyrin–phospholipids (CoPoPs) which can be co-formulated with lipid adjuvants, such as synthetic monophosphoryl lipid A (e.g., 3D6A-PHAD), and can transform soluble antigens into antigen particles by simple incubation, resulting in potent immune responses [31,32,33,34]. CoPoP liposomes induce the stable binding of His-tagged proteins and peptides due to the interaction of His-tag with cobalt in the lipid bilayer [35]. Previous studies have demonstrated that CoPoP liposomes can induce functional immune responses in mice and rabbits [36,37,38,39]. Moreover, the SARS-CoV-2 RBD is well-suited for particle formation with CoPoP liposomes to induce neutralizing antibody responses in both liquid [40] and lyophilized formulation [41]. Recently, this approach has entered human clinical trials with the EuCorVac-19 vaccine (ClinicalTrials.gov identifier NCT04783311).

SARS-CoV-2 structural proteins include S, envelope (E), membrane (M) and N. The S protein is responsible for viral binding, fusion, and entry into host cells, and contains an S1 region that is recognized by receptors on human cells and a conserved S2 region that mediates viral fusion to human cells [42]. Like several other receptor-binding proteins, SARS-CoV-2 S is a main vaccine target, and S binds to the human angiotensin-converting enzyme 2 (hACE2) receptor [43,44,45]. Most of the neutralizing monoclonal antibodies developed are against the RBD motif within S1 [46,47], and neutralizing antibodies generated during human SARS-CoV-2 infection correlate with antibodies against the RBD [48]. Therefore, the SARS-CoV-2 RBD is an intriguing target protein for vaccine development, and could potentially generate a higher percentage of specific neutralizing antibodies compared to the entire S immunogen. One of the issues of targeting the RBD is that mutations of SARS-CoV-2 variants often escape the neutralization of antibodies induced by RBD vaccine immunization [49], indicating that frequent vaccine updating or the targeting of other conserved S2 regions might be an approach in the future to combat variants. The present study presents a proof-of-concept study using CoPoP liposomes through I.N. immunization. K18 hACE2 mice were used, which express the human ACE2 receptor and are permissive to human SARS-CoV-2 lethal infection [50].

## 2. Results

The His-tagged RBD protein was admixed with CoPoP/3D6A-PHAD liposomes to generate a liposome-displayed RBD (abbreviated herein as “CP/RBD”). Vaccine-induced anti-RBD IgG and IgA titers in the lungs and sera in K18 hACE2 transgenic mice were then assessed. The mice were immunized, either intramuscularly (I.M.) or intranasally (I.N.), with CP/RBD at 2, 1, or 0.5 µg RBD on day 0 and 14 (Figure 1A). The 2 µg RBD dose also contained 8 µg of CoPoP and 3.2 µg of 3D6A-PHAD, and the 0.5 µg dose correspondingly contained a quarter of the dose of CoPoP and 3D6A-PHAD. We note that the 50 µL I.N. volume used resulted in vaccine delivery to the lower airway of the mice and, therefore, is not a faithful representation of human I.N. immunization. Following immunization, lung homogenates were collected on day 28. A higher level of anti-RBD IgA was observed in mice with I.N. immunization, compared to mice with I.M. immunization (Figure 1B), showing that I.N. immunization induced specific anti-RBD IgA titers within the lungs of K18 hACE2 transgenic vaccinated mice. In contrast, I.M. immunization did not induce detectable anti-RBD IgA titers in the lungs. Interestingly, both I.M. and I.N immunization induced anti-RBD IgG titer in lung homogenates (Figure 1C). We did not assess how much of this IgG was confined to lung blood vasculature and how much was secreted. The post-immunize sera were also collected to measure the anti-RBD IgG titer. A dose-dependent RBD-specific IgG titer in K18 hACE2 transgenic mice was observed when immunized with CP/RBD at 0.5, 1, or 2 µg RBD doses (Figure 1D). In most conditions, I.M. immunization induced a significantly higher anti-RBD IgG titer in the blood. This contrasts with I.N. immunization, which resulted in an elevated presence of anti-RBD IgA in the lungs. 

To access the functional antibodies induced by CP/RBD vaccination, a SARS-CoV-2 surrogate virus neutralization test (sVNT) was used. The post-immune sera and lung homogenates were diluted 100-fold. The lung homogenates from CP/RBD immunized mice with I.N. administration inhibited 96% and 90% of the interaction between RBD and ACE2 at a 2 and 1 µg RBD vaccine dose, respectively. On the other hand, post-immune lung homogenization in mice immunized through I.M. administration only inhibited 71% and 65% of the interaction between RBD and ACE2, at a 2 and 1 µg RBD dose (Figure 1E). The post-immune sera from I.N. administered mice inhibited 92% and 72% of the interaction between RBD and ACE2 at an administration dose of 2 and 1 µg RBD, with a 92% and 99% inhibition of the interaction between RBD and ACE2 from mice immunized through I.M. administration (Figure 1F). In this functional test, no statistical differences were detected, so more experiments would be needed with larger group sizes to confirm the trends. The sVNT results indicate that both I.M. and I.N. immunization induced functional antibodies in both the lungs and sera that could block the interaction between RBD and the ACE2 receptor. The impact of lung IgA relative to lung IgG was not discerned in this assay. 

Next, we conducted an IFNγ ELISPOT assay to detect antigen-specific T cells after post-immune spleen and lung cells were restimulated with the RBD antigen. The number of RBD-specific spot-forming cells (SFC) was assessed in K18 hACE2 transgenic mice immunized with liposomal RBD at a 2 µg antigen dose, either I.M. or I.N. The I.N. route resulted in more SFC being generated in the lungs compared to mice immunized I.M. (Figure 2A,B). This indicates that antigen-specific T cells were induced, and remained resident in the lungs following I.N. vaccination, when compared to I.M. On the other hand, mice immunized with an I.M. injection showed a higher frequency of SFC generated in the spleen after re-stimulation with the RBD antigen compared to I.N. immunized mice (Figure 2A lower panel, Figure 2B). 

To determine whether CP/RBD liposome immunization resulted in the uptake of the particles within immune cells in the lungs, we formulated CP liposomes with cobalt-free PoP (generating “CPP”), as a fluorescent tracer [51]. We assessed the resulting RBD particle uptake in APCs in the lungs 24 h after administration to K18 hACE2 transgenic mice. Following lung collection, resident APCs were examined for CPP liposome uptake by flow cytometry detecting PoP, and the surface markers CD11c (for dendritic cells) and F4/80 (for macrophages), as shown in Figure 3A and Figure 3B. This study showed that I.N. administration of the RBD displayed on CPP liposomes (CPP/RBD) resulted in particle uptake in both lung APCs at the 24 h time point. 

Next, K18 hACE2 transgenic mice were immunized I.M. or I.N. with 0.5 µg of liposome-displayed RBD. Each mouse was vaccinated with 0.5 µg of RBD (along with 2 µg of CoPoP and 0.8 µg of PHAD-3D6A) on day 0 and 14. On day 28, the immunized mice were challenged I.N. with a lethal dose of 10^5^ plaque-forming units (PFU) of SARS-CoV-2 WA-1 strain, as represented in Figure 4A. On day 2 and 4 after the viral challenge, one cohort was sacrificed, and viral loads in the nasal turbinate and lungs were assessed. The virus levels were significantly lower in CP/RBD-immunized mice compared to the control mice immunized with PBS (Figure 4B,C). Notably, immunized mice had undetectable levels of virus with I.M. delivery of CP/RBD on day 2 and 4 post-infection. The I.N. route was also effective at reducing viral loads relative to non-immunized animals. The suppression of viral loads in the vaccinated mice was further corroborated, with a 100% survival rate of both I.N. and I.M immunized mice, without significant weight loss (Figure 4D,E).

## 3. Discussion

I.N. immunization offers a convenient route for vaccination and also has the potential to stimulate mucosal immune responses. Virus infection could be blocked at the site of entry by specific antibodies in the respiratory tract. A limited number of nasal vaccines have been approved for human use, potentially due to the difficulty and variability in this route of administration. In this study, we note the limitation that with the injection volume used, most of the liposomes drained into the lungs and lower airways, which could be different from human I.N. vaccination. Nevertheless, several studies have indicated that I.N. vaccines could induce protection against SARS-CoV-2 in preclinical studies [52,53,54]. For example, in mice, an S subunit vaccine, combined with stimulator of interferon genes (STING) adjuvant, induced robust immunogenicity with a single I.N. injection [55]. An I.N. administration of Newcastle disease virus (NDV)-based vectored vaccine in mice and hamsters could induce immunogenicity against SARS-CoV-2 [56]. The I.N. administration of S RBD conjugated with a diphtheria toxoid induced a strong, localized immune response in the respiratory tract of K18-hACE2 transgenic mice, against a lethal challenge with SARS-CoV-2 [53]. Our approach is comparable to another study using K18-hACE2 transgenic mice administered 20 µg of RBD conjugated to a diphtheria toxoid (EcoCRM^®^), through either I.N. or I.M. routes, which also observed protection against a virus challenge (10^4^ PFU/animal) [53]. Our study provided complete protection, with a 40-fold lower RBD dose and a 10-fold higher viral challenge dose (10^5^ PFU/animal), but further experiments would be required to determine the relative efficacy amongst different vaccines. Overall, the present study adds to the growing body of evidence that the I.N. route is viable for SARS-CoV-2 vaccines, at least in a preclinical setting.

In this study, CP/RBD induced RBD-specific IgA in the lungs with I.N. but not I.M. immunization. On the other hand, I.M. immunization induced stronger antigen-specific IgG antibody levels in sera. Assessing the viral load following the challenge showed that the low level of IgA produced in the lungs was not sufficient to clear the virus in the lungs at early time points as effectively as I.M. immunization. Cellular immunity could play an important role in clearing SARS-CoV-2 infection [57,58]. A clinical study has shown that resident-specific T-cells in the lungs are correlated with survival after SARS-CoV-2 infection [59]. Here, we have shown that mice immunized I.N. induce RBD-specific T cells in the lungs, while I.N. immunization did not appear to induce as many systemic RBD-specific T cells, as judged by ELISpot results, in the spleen. These data are consistent with I.N. immunization inducing a localized antibody and a T cell immune response, whereas I.M immunization provides a more systemic response. One limitation is that the phenotype or function of the induced T cells was not further assessed in this study.

An I.N. vaccination has the advantage of mimicking the natural route of viral infection and eliciting mucosal antibody and cellular responses. Additional advantages of I.N. include needle-free administration and the potential opportunity for self-administration. The present study establishes a proof of principle for the CoPoP liposome platform to be further explored for I.N. administration. Future studies should assess functional vaccine dose–response and local toxicity effects, and should better emulate human-relevant I.N. immunization by using smaller vaccine volumes and testing the approach in larger animal models. Another interesting area of potential study could be to combine I.N. and I.M. immunization to maximally induce local and systemic immune responses. 

## 4. Conclusions

In summary, these data show that the I.N. administration of liposome-displayed RBD induces protection against a lethal challenge with SARS-CoV-2 in K18 hACE2 transgenic mice. I.N. vaccination also reduced viral burden in the lung and nasal turbinates following viral challenge, albeit not as well as I.M. administration. However, I.N. immunization elicited measurable anti-RBD IgA titers in the lungs, whereas the I.M did. The I.N. route appears viable for recombinant protein-based, liposomal SARS-CoV-2 vaccine candidates and could be optimized and investigated further in future studies. 

## 5. Experimental

**Materials:** His-tagged RBD (SARS-CoV02 Wuhan-Hu-1 S amino acids 319-541), expressed in human embryonic kidney 293 cells (HEK293), was acquired from RayBiotech (Peachtree Corners, GA, USA). PoP and CoPoP were produced as previously described [60]. The following lipids were used: 1,2-Dioleoyl-sn-glycero-3-phosphocholine (DOPC, Avanti Cat. # 850375), cholesterol (PhytoChol, Wilshire Technologies, Princeton, NJ, USA), synthetic monophosphoryl Hexa-acyl Lipid A, and 3-Deacyl (3D6A-PHAD, Avanti Cat # 699855P). CD11c APC (Clone: N418; Cat # 117310; Lot: B253461), and F4/80 PE (Clone: BM8; Cat # 123109; Lot: B251636) were used for flow cytometry and were obtained from Biolegend.

**Liposome preparation:** CoPoP- and PHAD-3D6A-containing liposomes (CP) were prepared by an ethanol injection method, followed by nitrogen-pressurized lipid extrusion in a phosphate-buffered saline (PBS) carried out at 55 °C [32]. The remaining ethanol was removed by dialysis against PBS twice at 4 °C. The final liposome concentration was adjusted to 320 µg mL^−1^ of CoPoP, and liposomes were passed through a 0.2 µm sterile filter and stored at 4 °C. The liposome size and polydispersity index were determined by dynamic light scattering (DLS) with a NanoBrook 90 plus PALS instrument, after 200-fold dilution in PBS. The CP liposome formulation had a mass ratio of [DOPC: CHOL: 3D6A-PHAD: CoPoP] [20:5:0.4:1]. The CPP liposomes included the non-metallic PoP as a fluorescent tracer and had a mass ratio of [DOPC: CHOL: 3D6A-PHAD: PoP: CoPoP] [20:5:0.4:0.6:0.4].

**Murine immunization and serum analysis:** All animal studies were carried out according to protocols approved by the University at Buffalo and Texas Biomedical Research Institute. Five-week-old K18 hACE2 transgenic mice (strain # 034860; Jackson Laboratories, Bar Harbor, ME, USA) received either I.M. or I.N. immunizations on days 0 and 14 containing 2, 1, and 0.5 µg RBD combined with CP liposomes. The vaccines were formulated in 50 µL and injected into the caudal thigh muscle for the I.M. route or administered I.N. to isoflurane-anesthetized mice being held upright by repeatedly applying ~10 µL of vaccine onto each nare until the vaccine was fully administered. The final bleeding was collected on day 28. The vaccines were prepared by incubating the RBD at a concentration of 80 µg mL^−1^ with liposomes (CoPoP or equivalent concentration of 320 µg mL^−1^) for 3 h at room temperature prior to dilution for immunization. 

**ELISA:** 1 µg mL^−1^ of RBD diluted in a coating buffer (28.5 mM Na_2_CO_3_; 71.4 mM NaHCO_3_, pH 9.6) was used to coat 96-well plates for 2 h at 37 °C. The wells were washed and then blocked with 2% BSA in PBS containing 0.1% Tween-20 (PBS-T) for 2 h at 37 °C. The mouse sera (serially diluted 10-fold in PBS-T containing 1% BSA) were incubated in the wells for 1 h at 37 °C, then washed with PBS-T. A goat anti-mouse IgG-HRP (Cat # A00160 from Genscript) was added. The wells were washed again with PBS-T before the addition of a tetramethylbenzidine (TMB) solution (Cat. # J60461 from Thermo Fisher Scientific). The antibody titers were defined as the reciprocal of serum dilution in which the absorbance at 450 nm exceeded the background by greater than 0.5 absorbance units. 

**RBD-hACE2 Inhibition Assay:** A SARS-CoV-2 cPass sVNT Kit (GenScript, Cat. L00847) was used to assess whether post-immune samples could block the interaction between hACE2 and an HRP –RBD antigen. The mouse sera and lung homogenates were diluted 100× with a sample dilution buffer. The positive and negative controls were included from the kit, and each of the control vials were diluted 10× with a sample dilution buffer. The diluted positive and negative controls, as well as the diluted samples, were mixed with an HRP–RBD solution at a 1:1 volume, then incubated at 37 °C for 30 min. After the incubation, 100 µL of the mixtures were transferred into each well of a pre-coated hACE2 ELISA plate and incubated at 37 °C for 15 min. The plate was washed 4 times to remove unbound HRP–RBD. The percentage of inhibition was calculated as % = (1 −  OD_450_ post immune sera/OD_450_ negative control) × 100%.

**Enzyme-linked immune absorbent spot (ELISpot) assay**: Splenocytes and lungs were harvested from immunized mice on day 28. The spleens were collected and passed through a 70 µm cell strainer in a 50 mL tube to collect single cells. The cells were centrifuged at 500 rcf, and a red blood lysis buffer was added for 5 min on ice to lyse red blood cells. After incubation, 20 mL of PBS was added to dilute the lysis buffer, and the samples were centrifuged at 500 rcf for 5 min. The lungs were collected and cut into pieces with scissors, then were digested with 3 mg/ml of collagenase I for 45 min at 37 °C, before filtering through a 70 µm cell strainer in a 50 mL tube to collect single cells. The cells were centrifuged at 500 rcf, and a red blood lysis buffer was added for 5 min on ice to lyse red blood cells. After incubation, 20 mL of PBS was added to dilute the lysis buffer, and samples were centrifuged at 500 rcf for 5 min.

A total of 3 × 10^5^ splenocytes or lung cells, were seeded in an ELISpot plate, and 5 µg/mL of RBD was added to each well. The cells were cultured in 5% CO_2_/95% air at 37 °C in a humidified chamber for 24 h. The detection of spots was performed according to the manufacturer instructions from Immunospot, using the Murine IFN-γ Single-Color ELISPOT kit. The next day, the plate was washed twice with PBS and twice with PBS-T. The wells were incubated with 80 µL of anti-murine IFNγ (Biotin) antibody detection buffer for 2 h. Later, the wells were washed three times with PBS-T, then incubated with Streptavidin-AP for 0.5 h. After the incubation, each well was washed with PBS-T twice and distilled water twice. To develop spots, the plates were incubated for 15 min at RT, with 80 μL per well of blue developer solution provided from the manufacturer. The images were acquired with CTL ImmunoSpot S6 FluoroCore analyzer.

**Liposome uptake study:** Mice received 1 µg of RBD displayed on liposomes containing CoPoP, PHAD-3D6A and PoP (as a fluorescent tracer) through I.N. administration. After 48 h, the mice were sacrificed and the lungs were collected. The lungs were dissected into pieces and incubated with 3 mg/ml of collagenase I in RPMI medium for 30 min at 37 °C. The lungs were collected and passed through a 70 µm cell strainer in a 50 mL tube to collect single cells. The cells were centrifuged at 500 rcf and a red blood lysis buffer was added and incubated for 5 min on ice. Later, 5 × 10^5^ cells per tube were strained then stained with murine antibodies against CD11c and F4/80 for 30 min on ice. The samples were washed twice with a FACS buffer (cold PBS containing 0.5% BSA and 0.05% sodium azide), prior to BD LSRFortessa TM X-20 flow cytometry. Flowjo (version 10) software was used for data analysis.

**Murine virus challenge:** Six- to eight-week-old female K18-hACE2 transgenic mice were acquired from Jackson Laboratories and maintained in micro-isolator cages in the Animal Biosafety Laboratory level 3 (ABSL3) at the Texas Biomedical Research Institute. The mice were provided with sterile water and chow ad libitum, and were acclimated for one week upon arrival before vaccination and challenge experiments. The mice were anesthetized using isoflurane and vaccinated intramuscularly or intranasally with PBS or 0.5 µg of RBD combined with CP liposomes on day 0 and day 14. At 28 days, after a booster vaccination, K18 hACE2 transgenic unvaccinated or vaccinated mice were challenged I.N. with a lethal dose (10^5^ PFU/mouse) of SARS-CoV-2, USA-WA1/2020 (WA-1) strain and monitored daily for morbidity (body weight) and mortality (survival). The mice that had lost more than 25% of their initial body weight were considered to have reached their experimental endpoint and were humanely euthanized. Mock-challenged K18 hACE2 transgenic mice were also included as controls. Concurrently, mice (n=3/group) were similarly vaccinated, infected, and euthanized on days 2 and 4 post-challenge to evaluate viral load in the nasal turbinate and lungs. The organs were homogenized in 1 ml of PBS using a Precellys tissue homogenizer (Bertin Instruments) and tissue homogenates were centrifuged at 21,500× *g* for 10 min. The tissue culture supernatants were collected, and viral titers were determined by standard plaque assay in Vero E6 cells.

## Figures and Tables

**Figure 1 pathogens-11-01035-f001:**
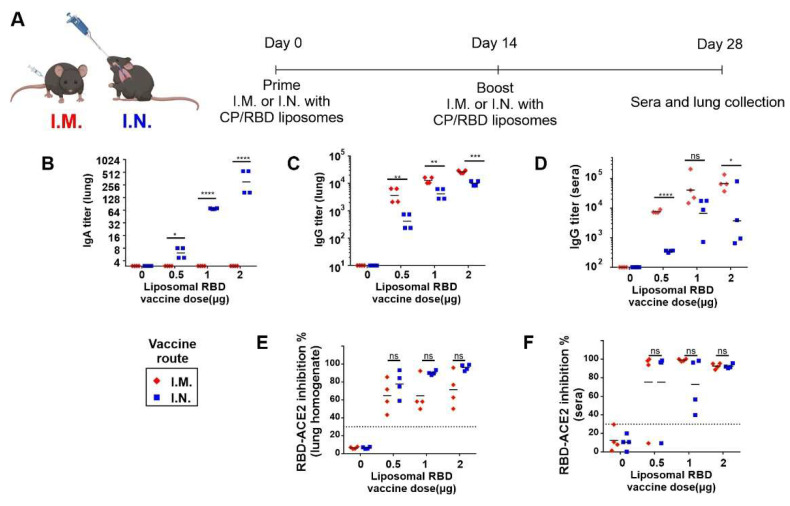
Immunogenic liposome-displayed RBD for I.M. and I.N. immunization. K18 hACE2 transgenic mice were immunized with CP/RBD on day 0 and 14 intranasally (I.N.; blue) or intramuscularly (I.M.; red). Lung homogenates and serum were collected on day 28. (**A**) Schematic representation of the immunization schedule. Anti-RBD lung IgA (**B**), lung IgG (**C**) and sera IgG (**D**) ELISA titers in K18 hACE2 transgenic mice immunized I.N. or I.M. with the indicated RBD dose. Antibody function was assessed with a sVNT assay with post-immune lung homogenates (**E**) and sera (**F**). *n* = 4 mice per group. Lines represent geometric (titer) and arithmetic (sVNT) mean. A two-sided Student T-test using log-transformed titer or sVNT data was used to analyze differences, * *p* < 0.05, ** *p* < 0.01, *** *p* < 0.001, **** *p* < 0.0001.

**Figure 2 pathogens-11-01035-f002:**
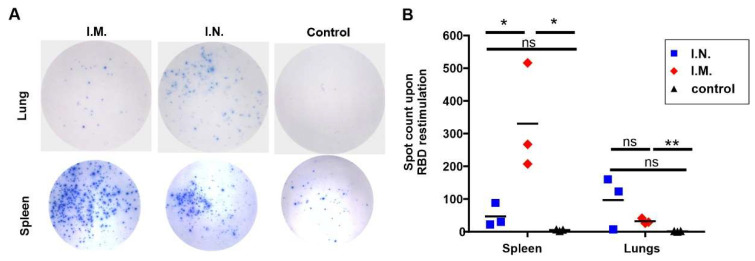
Cellular responses induced by CP/RBD immunization. K18 hACE2 transgenic mice were immunized with CP/RBD on day 0 and 14. Lung and splenocytes were collected and restimulated with the RBD. (**A**) Images of ELISpot results from lungs (top) and spleen (bottom). (**B**) Quantification of results shown in (**A**). The lines in (**B**) represent mean, and two-sided Student T-test using log-transformed data was used to analyze differences, * *p* < 0.05, ** *p* < 0.01. Lung and spleen cells from *n* = 5 mice per group were pooled, and the samples were performed in triplicate.

**Figure 3 pathogens-11-01035-f003:**
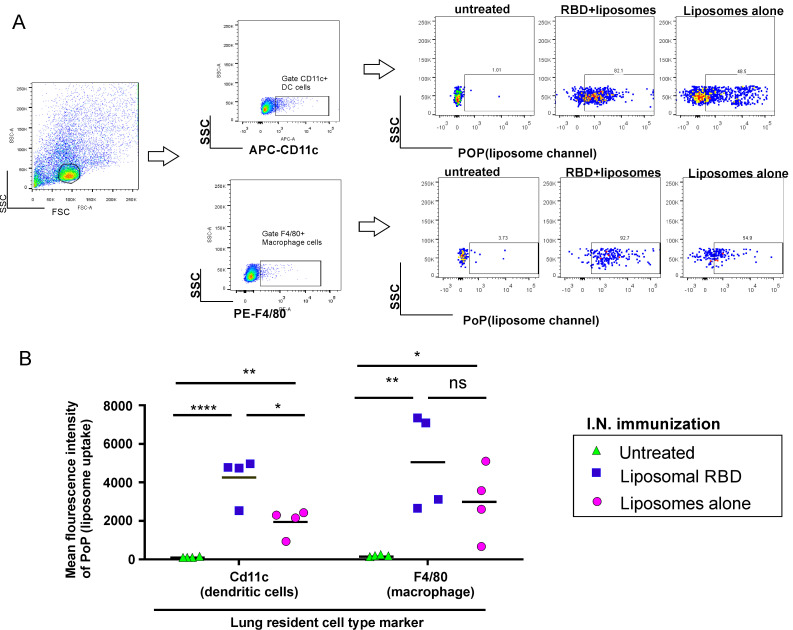
Uptake of antigen particles into immune cells of the lungs. K18 hACE2 transgenic mice were I.N. administered liposomes also containing CoPoP, PoP (for fluorescence detection) and 3D6A-PHAD with or without RBD display. Two days later, lungs were collected, and liposomal uptake was assessed with flow cytometry. (**A**) Dendritic cells were gated with CD11c-positive cells and macrophages were gated with F4/80-positive cells. (**B**) The number of cells with PoP fluorescence in indicated cell type. *n* = 4 mice per group. The lines in **C** represent mean, and two-sided Student T-test using log-transformed data was used to analyze differences, * *p* < 0.05, ** *p* < 0.01, **** *p* < 0.0001.

**Figure 4 pathogens-11-01035-f004:**
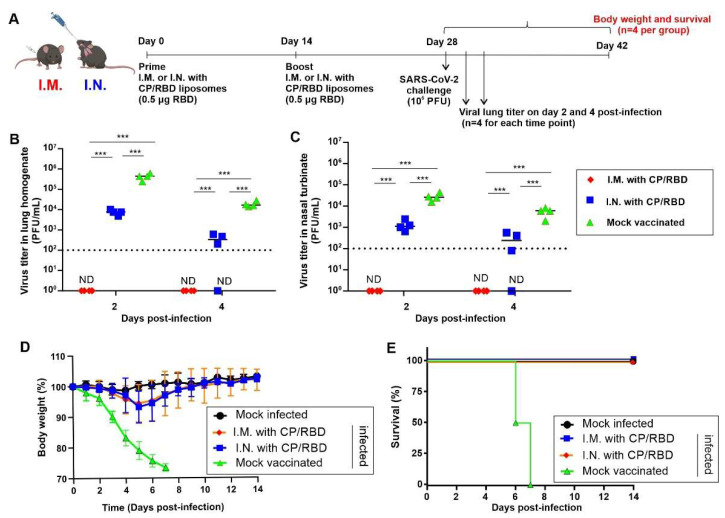
Protection of K18-hACE2 transgenic mice against lethal SARS-CoV-2 challenge following I.N. or I.M. vaccination with CP/RBD. K18-hACE2 transgenic mice (*n* = 12 per group) were immunized I.M. or I.N. with CP/RBD (0.5 µg RBD) on days 0 and 14 prior to challenge on day 28 with 10^5^ PFU of SARS-CoV-2 WA-1. Mock-vaccinated and mock-infected K18-hACE2 transgenic mice were included as controls. (**A**) Schematic representation of the immunization schedule and challenge. On day 2 and day 4 post-infection, a cohort of 4 mice was euthanized, lungs (**B**) and nasal turbinates (**C**) were collected and homogenized, and viral titers were determined by standard plaque assay. The dotted line shows the limit of detection of the assay. “ND” is not detected. Mice were monitored post-challenge for 14 days for body weight loss (**D**) and survival (**E**). Student t-test was used to determine significance between viral titers, *** *p* < 0.005.

## Data Availability

All raw data are available upon request.

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
