# Peer review of "Intranasal Immunization with Liposome-Displayed Receptor-Binding Domain Induces Mucosal Immunity and Protection against SARS-CoV-2"

_pathogens, 2022, doi:10.3390/pathogens11091035_

Round 1
Reviewer 1 Report
There are a few major issues that the authors have to address.
1. The presented data does not support that Intranasal immunization with a robust local IgA response might lead to a better protection against infection compared to IM. Please provide a reasonable conclusion and discussion.
2. For figure 1, the authors show the IgA level in the lung, however, it is not providing any correlation to protection. You have to do the neutralization testing for the lung homogenates.
3. For figure 2, please discuss the possible effects of low SFC in spleen with IN immunization.
4. Figure 4A, No x title. Figure 4B, what does the time mean? There are apparently high level of virus load in the lung from IN compared to IM. The slow clearance of virus will increase risks of virus mutation to escape. Please justify what is the advantage of this vaccine for future application. Based on the data provided, there is no sense to use this vaccine to replace IM vaccines.
Author Response
Response to Reviewer 1
There are a few major issues that the authors have to address.
- The presented data does not support that Intranasal immunization with a robust local IgA response might lead to a better protection against infection compared to IM. Please provide a reasonable conclusion and discussion.
Author response: We agree with the referee about this matter. We have tried to clarify that results presented in the manuscript are a proof-of–concept that liposomal vaccine platform is viable for IN immunization, and leads to protection against an otherwise lethal disease challenge using sub microgram vaccine doses. As there is still a great interest in SARS-CoV-2 vaccines, we think this is notable. We have clarified in the abstract, results and discussion of the revised manuscript that IM provides a better level of viral clearance as compared to IN immunization. In the revised conclusion section we have stated “these data show that the IN route is viable for liposomal-based SARS-CoV-2 vaccine candidates that can be optimized and investigated further in future studies.”
- For figure 1, the authors show the IgA level in the lung, however, it is not providing any correlation to protection. You have to do the neutralization testing for the lung homogenates.
Author response: We thank the reviewer for the suggestions. Following the recommendation made b the reviewer, we have performed a surrogate virus neutralization test for the lung homogenates and have included this new results in the revised Fig 1E and F.
- For figure 2, please discuss the possible effects of low SFC in spleen with IN immunization.
Author response: Following the suggestion made by the reviewer, we have added the following text in the discussion section of the revised document:
Here, we have shown that mice immunized IN induce RBD-specific T cells in lung while IN immunization did not appear to induce as many systemic RBD-specific T cells as judged by ELISpot results in the spleen. This data is consistent with IN immunization inducing a localized antibody and T cell immune response whereas IM immunization provides a more systemic response. One limitation is that the phenotype or function of the induced T cells was not further assessed in this study.
- Figure 4A, No x title. Figure 4B, what does the time mean? There are apparently high level of virus load in the lung from IN compared to IM. The slow clearance of virus will increase risks of virus mutation to escape. Please justify what is the advantage of this vaccine for future application. Based on the data provided, there is no sense to use this vaccine to replace IM vaccines.
Author response: We have clarified the issues in Figure 4. We agree with the referee's interpretation and have tried to make that clear in describing the results. However, the detection of localized IgA and cellular responses in the lung of IN immunized mice is noteworthy, as is the full protection observed. As stated above, it is possible, although currently outside the scope of this study, that with different dosing regimens, IN immunization could be further improved. We have included the following sentence in the revised discussion of the manuscript: "Future studies should also assess functional vaccine dose responses, local toxicity effects, and should better emulate human-relevant I.N. immunization by using smaller vaccine volumes and testing the approach in different animal models. Another interesting area of potential study could be to combine I.M. and I.N. immunization to maximally induce local and systemic immune responses."
Reviewer 2 Report
The manuscript discussed IN RBD vaccination using liposome particles as was developed in the previous manuscripts.
The manuscript is well written, methods and results are well discussed.
Several minor suggestions including:
1. adding illustration on how the experiments were done, so it will be easier to follow.
2. giving more explanation or reference for "standard" / common methods used.
3. Some markings needs to be added to the graph of survival, on which 2 groups marking was not added
4. Minor typhological errors were found.
Author Response
Response to Reviewer 2
The manuscript discussed IN RBD vaccination using liposome particles as was developed in the previous manuscripts. The manuscript is well written, methods and results are well discussed. Several minor suggestions including:
- Adding illustration on how the experiments were done, so it will be easier to follow.
Author response: We thank the referee for this suggestion. Following the recommendation made by the reviewer, we have added schematic illustrations for the experiments in Fig 1 and Fig 4.
- Giving more explanation or reference for "standard" / common methods used.
Author response: We have substantially expanded the methods section in the revised version of the manuscript.
- Some markings need to be added to the graph of survival, on which 2 groups marking was not added
Author response: We apologize for not providing this information in the original submission of the manuscript and we thank the referee for the suggestion. This has been corrected in the revised Figures of the new submission.
- Minor typological errors were found.
Author response: We apologize for these editorial deficiencies. We have gone through the manuscript and corrected the errors we could find.
Reviewer 3 Report
This paper by Chiao Huang et al. investigates the efficacy of a liposome vaccine for SARS-CoV-2 and compares intranasal and intramuscular administration in a K18 human ACE2 transgenic mouse model. In this model, intranasal administration resulted in more specific IgA, but less specific IgG in the lung. Intramuscular immunization induced a larger T cell ELISpot response in the spleen compared to intranasal immunization. When immunized mice are challenged, survival rates are similar for both administration routes. The methods used in this paper are sufficient to answer the questions posed, although numbers are small.
Comments:
The introduction implies that current SARS-CoV-2 vaccines do not induce a mucosal response, while there are multiple papers reporting the detection of mucosal (nasal) SARS-CoV-2 IgA antibodies after immunization. Please adjust.
Figure 4 does not match the text as figure 4E, referenced in the text is not present in the figure. In addition no statistics are present for this figure even though statistical differences are mentioned in the text.
In addition, it seems that in figure 4B, two out of four mock infected mice died, while 100% survival is observed for the other groups. This does not match figure 4A. Please explain.
The current paper introduces two “new” concepts compared to the SARS-CoV-2 vaccines that are currently in use, namely a liposome vaccine and intranasal administration. However the main focus is on a comparison between intranasal and intramuscular administration. How does the liposomal vaccine described here compare to other vaccines currently in use with regards to (immunological) performance in the K18-hACE2 mouse model? If it is not possible to provide a direct comparison, data from other studies should be discussed and compared.
Author Response
Response to Reviewer 3
The introduction implies that current SARS-CoV-2 vaccines do not induce a mucosal response, while there are multiple papers reporting the detection of mucosal (nasal) SARS-CoV-2 IgA antibodies after immunization. Please adjust.
Author response: Thank you for pointing this out. We have added the following text in the revised introduction section of the manuscript:
To date, approved vaccines for coronavirus disease 2019 (COVID-19) and most vaccines in development are administered through the intramuscular (I.M.) route, which produce systemic immune responses including serum IgG, and it is remains a matter of controversy whether these produce meaningful mucosal immune responses [17]. SARS-CoV-2 mRNA-based vaccines, administered systemically, induce only weak mucosal immune responses [18].
- Su, F.; Patel, G.B.; Hu, S.; Chen, W. Induction of mucosal immunity through systemic immunization: Phantom or reality? Hum Vaccin Immunother 2016, 12, 1070-1079, doi:10.1080/21645515.2015.1114195.
- Tang, J.; Zeng, C.; Cox, T.M.; Li, C.; Son, Y.M.; Cheon, I.S.; Wu, Y.; Behl, S.; Taylor, J.J.; Chakraborty, R., et al. Respiratory mucosal immunity against SARS-CoV-2 following mRNA vaccination. Sci Immunol 2022, 10.1126/sciimmunol.add4853, eadd4853, doi:10.1126/sciimmunol.add4853.
Figure 4 does not match the text as figure 4E, referenced in the text is not present in the figure. In addition no statistics are present for this figure even though statistical differences are mentioned in the text.
In addition, it seems that in figure 4B, two out of four mock infected mice died, while 100% survival is observed for the other groups. This does not match figure 4A. Please explain.
Author response: We apologize for these editorial mistakes that haven been corrected in the revised version of the manuscript. Following the comment made by the reviewer, we have added statistics and corrected the other errors mentioned.
The current paper introduces two “new” concepts compared to the SARS-CoV-2 vaccines that are currently in use, namely a liposome vaccine and intranasal administration. However the main focus is on a comparison between intranasal and intramuscular administration. How does the liposomal vaccine described here compare to other vaccines currently in use with regards to (immunological) performance in the K18-hACE2 mouse model? If it is not possible to provide a direct comparison, data from other studies should be discussed and compared.
Author response: We thank the reviewer for bringing this important point to our attention. We compared our vaccine with other adjuvant using the K18-hACE2 mice through I.N. administration in the discussion section of the revised manuscript:
Nevertheless, several studies have indicated that I.N. vaccines could induce protection against SARS-CoV-2 in pre-clinical studies [52-54]. For example, in mice, a S subunit vaccine combined with stimulator of interferon genes (STING) adjuvant induced robust immunogenicity with a single I.N. injection [55]. I.N. administration of Newcastle disease virus (NDV)-based vectored-vaccine in mice and hamsters could induced immunogenicity against SARS-CoV-2 [56]. I.N. administration of S RBD conjugated with Diphtheria toxoid induced a strong localized immune response in the respiratory track in K18-hACE2 transgenic mice against lethal challenge with SARS-CoV-2 [53]. Our approach is comparable to another study using K18-hACE2 transgenic mice administrated with 20 µg RBD conjugated to Diphtheria toxoid (EcoCRM®) through either I.N. or I.M. routes, which also observed protection against virus challenge (104 PFU/animal) [53]. Our study provided complete protection with a 40-fold lower RBD dose and a 10-fold higher viral challenge dose (105 PFU/animal), but further experiments would be required to determine relative efficacy amongs different vaccines. Overall the present study adds to this current growing body of evidence that the I.N. route is viable for SARS-CoV-2 vaccines, at least in a preclinical setting.
- Afkhami, S.; D'Agostino, M.R.; Zhang, A.; Stacey, H.D.; Marzok, A.; Kang, A.; Singh, R.; Bavananthasivam, J.; Ye, G.; Luo, X., et al. Respiratory mucosal delivery of next-generation COVID-19 vaccine provides robust protection against both ancestral and variant strains of SARS-CoV-2. Cell 2022, 185, 896-915 e819, doi:10.1016/j.cell.2022.02.005.
- Wong, T.Y.; Lee, K.S.; Russ, B.P.; Horspool, A.M.; Kang, J.; Winters, M.T.; Allison Wolf, M.; Rader, N.A.; Miller, O.A.; Shiflett, M., et al. Intranasal administration of BReC-CoV-2 COVID-19 vaccine protects K18-hACE2 mice against lethal SARS-CoV-2 challenge. NPJ Vaccines 2022, 7, 36, doi:10.1038/s41541-022-00451-7.
- van Doremalen, N.; Purushotham, J.N.; Schulz, J.E.; Holbrook, M.G.; Bushmaker, T.; Carmody, A.; Port, J.R.; Yinda, C.K.; Okumura, A.; Saturday, G., et al. Intranasal ChAdOx1 nCoV-19/AZD1222 vaccination reduces viral shedding after SARS-CoV-2 D614G challenge in preclinical models. Sci Transl Med 2021, 13, doi:10.1126/scitranslmed.abh0755.
- An, X.; Martinez-Paniagua, M.; Rezvan, A.; Fathi, M.; Singh, S.; Biswas, S.; Pourpak, M.; Yee, C.; Liu, X.; Varadarajan, N. Single-dose intranasal vaccination elicits systemic and mucosal immunity against SARS-CoV-2. bioRxiv 2020, 10.1101/2020.07.23.212357, doi:10.1101/2020.07.23.212357.
- Park, J.G.; Oladunni, F.S.; Rohaim, M.A.; Whittingham-Dowd, J.; Tollitt, J.; Hodges, M.D.J.; Fathallah, N.; Assas, M.B.; Alhazmi, W.; Almilaibary, A., et al. Immunogenicity and protective efficacy of an intranasal live-attenuated vaccine against SARS-CoV-2. iScience 2021, 24, 102941, doi:10.1016/j.isci.2021.102941.
Round 2
Reviewer 1 Report
Good work.
Reviewer 3 Report
The authors have answered my questions and improved the paper sufficiently.